# Beneficial effects of climate warming on boreal tree growth may be transitory

Loïc D'Orangeville [1,6], Daniel Houle[2,3], Louis Duchesne[2], Richard P. Phillips[4]
Yves Bergeron [1,5] & Daniel Kneeshaw[1]

Predicted increases in temperature and aridity across the boreal forest region have the potential to alter timber supply and carbon sequestration. Given the widely-observed variation in species sensitivity to climate, there is an urgent need to develop species-specific predictive models that can account for local conditions. Here, we matched the growth of 270,000 trees across a 761,100 km$^2$ region with detailed site-level data to quantify the growth responses of the seven most common boreal tree species in Eastern Canada to changes in climate. Accounting for spatially-explicit species-specific responses, we find that while 2 °C of warming may increase overall forest productivity by 13 ± 3% (mean ± SE) in the absence of disturbance, additional warming could reverse this trend and lead to substantial declines exacerbated by reductions in water availability. Our results confirm the transitory nature of warming-induced growth benefits in the boreal forest and highlight the vulnerability of the ecosystem to excess warming and drying.

[1] Centre for Forest Research, Université du Québec à Montréal, Case Postale 8888, Succ. Centre-Ville, Montreal, QC H3C 3P8, Canada. [2] Direction de la Recherche Forestière, Ministère des Forêts, de la Faune et des Parcs du Québec, 2700 Einstein, Quebec City, QC G1P 3W8, Canada. [3] Ouranos, 550 Rue Sherbrooke O, Montréal, QC H3A 1B9, Canada. [4] Department of Biology, Indiana University, 1001 East 3rd Street, Bloomington, IN 47405-7005, USA. [5] NSERC-UQAT-UQAM Industrial Chair in Sustainable Forest Management, Forest Research Institute, Université du Québec en Abitibi-Témiscamingue, 445 de l'Université, Rouyn-Noranda, QC J9X 5E4, Canada. [6] Present address: Faculty of Forestry and Environmental Sciences, University of New Brunswick, 28 Dineen Drive, Fredericton, NB E3B 5A3, Canada. Correspondence and requests for materials should be addressed to L.D'O. (email: loicdorangeville@gmail.com)

Climate models predict increases in temperature and aridity across the boreal forest region[1] that are likely to exceed 2 °C by the end of the century[2,3]. While warmer and drier conditions are typically thought to reduce tree growth[4–8], rising temperature and aridity may increase growth in cool, wet boreal regions where excess water can hinder forest productivity throughout much of the growing season[9–11]. Eastern North America (ENA) is projected to be the planet's only boreal region with sufficient precipitation to cancel out increases in evapo-transpiration associated with future warming[12]. Given the high degree of inter- and intra-specific variation in climate sensitivity of boreal species to recent warming[9,13,14], we still lack accurate esti-mates of species climatic thresholds. Such spatially-explicit models that account for site characteristics and local heterogeneity in temperature and soil water are urgently needed to predict future trajectories for this ecosystem and inform management strategies, global climate models, and climate-change mitigation actions[15,16].

Although climate envelope models provide insights into species' adaptive capacity[17], they display inconsistent responses and lagged sensitivity to climate change[18]. Radial growth has strong connec-tions to vital forest demographic rates including tree mortality and fecundity[19–22], while relationships between growth and environ-mental drivers yield information on a species' adaptive capacity[20]. In this sense, demographic performance indices may provide higher-resolution information on species adaptation to changing climate. Furthermore, models that can account for non-linear growth responses are needed to detect climatic thresholds beyond which climate effects may shift from positive to negative[23]. Building such models can only be achieved based on numerous observations over a large range of climatic conditions.

Here, we aimed to model sensitivity to current climate of the seven most abundant boreal tree species in Eastern Canada (black spruce, *Picea mariana*; white spruce, *Picea glauca*; balsam fir, *Abies balsamea*; jack pine, *Pinus banksiana*; aspen, *Populus tremuloides*; white birch, *Betula papyrifera*; and Larch, *Larix laricina*) and to project potential changes in growth to increasing temperature and changes in precipitation. First, general additive models (GAM) were used to model the growth (1985–2005) of 270,000 trees across 95,000 temperate and boreal stands, while accounting for local climate, tree size and age, soil characteristics, successional stage, and competition with neighboring trees. Second, models were used to assess the local vulnerability of 141 million inventoried stems in the boreal zone under study to an array of temperature and pre-cipitation change scenarios based on a suite of general circulation models (GCM). The fact that our models were fitted to growth observations extending to the warmer temperate zone allowed us to simulate boreal growth responses to warming while remaining within the observed climate space.

Our study presents striking species-specific variation in climate sensitivity across the study area. The growth of most conifers is mostly limited by water scarcity in southern regions but con-strained by low temperatures in northern regions. To the contrary, birch and aspen appear less vulnerable in the southern range of their distribution. In the absence of disturbance, the sum of pro-jected southern declines and northern increases in growth across the boreal zone suggests net growth gains with warming up to 2 °C. Additional warming reverses this trend, leading to growth declines exacerbated by reductions in water availability. Such results high-light the limited capacity of boreal forests in ENA to adapt to future climate change, which hinges on hypothetical increases in precipitation.

## Results

**Growth models.** Overall, stand characteristics and climate were strong predictors of tree growth for all seven species across the study area (Fig. 1), explaining between 52 and 70% of the deviation (Table 1). Relative mean square error was small (range: 5.2–6.1%), indicating sufficient sample size, residuals were normal (Supplementary Fig. 1) and cross-validation revealed good pre-dictive capacity (Table 1). Despite large inter-species differences in growth rates, effects of growth drivers were generally consistent across species. Growth increased exponentially with tree size ($P <$ 0.001; Student t-test) while growth rates declined sharply with age in young trees (ca. <50 years; Table 1 and Supplementary Fig. 3). Both symmetric (BA) and asymmetric competition (BAL) had significant ($P < 0.001$; Wald test) negative effects on tree growth in all species (Table 1 and Supplementary Fig. 3). Growth of spruce species was higher on sites with low to moderate terrain slope (0–20%), while steep slopes negatively affect the growth of aspen and balsam fir ($P < 0.05$; Wald test).

Growth responded strongly to mean daily maximum tempera-ture ($T_{MAX}$), growing season water availability (climate moisture index (CMI), measured as growing season precipitation minus potential evapotranspiration (PET), see Methods), and the interaction of the two ($P < 0.01$; Wald test; Table 1). The one

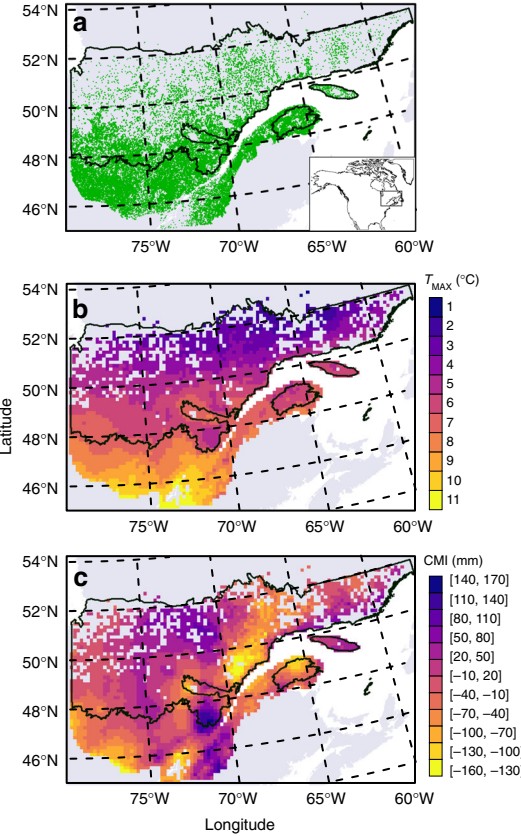

**Fig. 1** Plot location and average climate of the study area. **a** Location of sampled plots (green) in Quebec, Canada (gray). **b** Average annual daily maximum temperature ($T_{MAX}$) across sampled plots. **c** Growing season (May to September) climate moisture index (CMI, see Methods). The intermediate black line indicates the limit between the boreal and temperate vegetation zones while the upper black line represents the limit for commercial forestry. Variables in **b**, **c** are averaged over the study period (1985–2005) and per 15-km polygon. Data for base maps from https://www12.statcan.gc.ca/census-recensement/2011/geo/bound-limit/bound-limit-2011-eng.cfm with permission under http://open.canada.ca/en/open-government-licence-Canada and from https://www.donneesquebec.ca/recherche/fr/dataset/systeme-hierarchique-de-classification-ecologique-du-territoire used with permission under a Creative Commons 4.0—Attribution CC BY

**Table 1 Environmental characteristics and model summary**

|  | White birch | White spruce | Black spruce | Larch | Aspen | Jack pine | Balsam fir |
|---|---|---|---|---|---|---|---|
| *Distribution* | | | | | | | |
| Plots | 20,338 | 9940 | 33,819 | 1054 | 10,777 | 5148 | 42,719 |
| Trees | 37,526 | 15,262 | 92,811 | 1946 | 21,905 | 12,068 | 89,097 |
| *Median values (5th, 95th percentiles)* | | | | | | | |
| BAI (cm$^2$ year$^{-1}$) | 4.5 (1.4, 12.3) | 9.2 (2.1, 27.3) | 2.9 (0.6, 9.1) | 6.5 (1.6, 21.3) | 8.3 (2.5, 24.8) | 4.3 (0.8, 12.9) | 5.8 (1.5, 17.2) |
| DBH$^a$ (cm) | 15 (8, 29) | 19 (7, 35) | 14 (8, 24) | 13 (5, 28) | 18 (6, 36) | 15 (6, 28) | 13 (6, 24) |
| Tree age$^a$ (years) | 50 (19, 99) | 54 (15, 121) | 67 (23, 171) | 35 (12, 104) | 42 (13, 82) | 46 (12, 95) | 43 (14, 105) |
| BA (m$^2$ ha$^{-1}$) | 26 (12, 63) | 29 (12, 76) | 25 (10, 57) | 24 (8, 59) | 28 (12, 56) | 24 (8, 61) | 30 (13, 69) |
| BAL (m$^2$ ha$^{-1}$) | 13 (3, 35) | 14 (3, 45) | 9 (1, 30) | 10 (1, 31) | 12 (3, 29) | 9 (1, 30) | 17 (4, 44) |
| Slope (%) | 9 (0, 31) | 9 (0, 31) | 5 (0, 26) | 0 (0, 12) | 9 (0, 26) | 4 (0, 21) | 9 (0, 31) |
| $T_{MAX}$ (°C) | 7.6 (6.1, 9.5) | 7.7 (5.9, 10.0) | 6.6 (3.5, 8.7) | 7.9 (5.6, 10.6) | 7.8 (6.3, 10.1) | 7.0 (4.5, 9.2) | 7.4 (4.7, 9.8) |
| CMI (mm) | −6 (−79, 116) | −20 (−92, 95) | 7 (−81, 83) | 3 (−76, 72) | −17 (−94, 60) | 6 (−72, 66) | 1 (−86, 145) |
| Snowfall (mm) | 159 (128, 236) | 171 (129, 245) | 154 (113, 222) | 151 (112, 208) | 153 (114, 215) | 144 (113, 166) | 180 (131, 251) |
| *Model fit (%)* | | | | | | | |
| Dev. Expl | 52.0 | 52.9 | 57.0 | 67.1 | 64.2 | 70.3 | 55.5 |
| RMSE | 5.4 | 6.0 | 5.6 | 5.2 | 5.6 | 5.8 | 6.1 |
| Test set RMSE | 6.9 | 8.8 | 5.7 | 5.9 | 7.5 | 8.3 | 6.6 |
| *Parametric coefficients (±SE)* | | | | | | | |
| Intercept | −3.2 ± 0.1*** | n.s. | n.s. | −2.4 ± 0.3*** | n.s. | n.s. | n.s. |
| log(DBH) | 0.93 ± 0.02*** | 1.06 ± 0.03*** | 0.37 ± 0.02*** | 0.78 ± 0.07*** | 1.20 ± 0.02*** | 0.72 ± 0.03*** | 0.65 ± 0.01*** |
| DBH (×10$^3$) | 1.8 ± 0.1*** | 1.3 ± 0.2*** | 5.0 ± 0.1*** | 4.0 ± 0.5*** | 2.2 ± 0.1*** | 3.4 ± 0.2*** | 4.0 ± 0.1*** |
| *Significance of smoothing terms* | | | | | | | |
| Age | *** | *** | *** | *** | *** | *** | *** |
| Slope | n.s. | *** | *** | n.s. | *** | n.s. | *** |
| BA | *** | *** | *** | *** | *** | *** | *** |
| BAL | *** | *** | *** | *** | *** | * | *** |
| $T_{MAX}$ | *** | *** | *** | ** | * | *** | *** |
| CMI | *** | ** | *** | n.s. | *** | ** | *** |
| $T_{MAX}$ * CMI | n.s. | n.s. | *** | * | n.s. | *** | *** |

Distribution and characteristics of sampled trees and plots, as well as predictive capacity, error and significance of parametric coefficients, and smoothed terms for each species growth model. The predictive capacity of each model was computed from models fit on a subset of data (80%) to predict growth of the remaining 20% of trees (i.e. test set)
BAI: basal area increment, DBH: diameter at breast height, BA: symmetric competition, BAL: asymmetric competition, $T_{MAX}$: mean annual maximum temperature, CMI: climate moisture index (May–September) (snowfall is the sum of January–March snowfall, in mm), Dev. Expl.: deviance explained, RMSE root mean square error, n.s.: non-significant
*$P < 0.05$ (Wald test)
**$P < 0.01$
***$P < 0.001$
$^a$DBH and tree age are averaged over each tree's growth period

exception was larch, which was relatively insensitive to available water and was therefore excluded from further analyses. This lack of response to water availability is likely due to the species predominance in wetlands (51% of sampled individuals; Supplementary Fig. 2).

We identified the non-linear effects of single climatic variables in mediating specific growth by holding remaining variables constant in the model (at their median value). The growth of white spruce and balsam fir increased up to $T_{MAX}$ values of 7.3 and 8.1 °C, respectively, but declined above these thresholds. Thresholds of 8.4 and 8.7 °C were observed for black spruce and white birch, respectively, although both species appear less sensitive to $T_{MAX}$ (Supplementary Fig. 3). Broadleaved species displayed a parabolic growth in response to CMI, with reduced growth rates at both ends of the gradient (Supplementary Fig. 3). Low moisture negatively impacted balsam fir growth while high moisture was associated with growth reductions in jack pine and spruce species. Only two species (aspen and jack pine) responded linearly (both positively) to warming and only two species (black spruce and jack pine) responded linearly (both negatively) to moisture across the climate gradient. Such linear relationships between growth and climate are probably due to a lack of samples from the warm end of the climatic gradient.

Importantly, temperature effects on growth were conditioned by water availability. On cold sites ($T_{MAX} < 6$ °C), the growth of northern conifers like jack pine and black spruce tended to decline with excess moisture (Fig. 2). On warm sites ($T_{MAX} >$ 8 °C), most species displayed water limitation. For balsam fir and jack pine, higher moisture minimized or canceled the growth decline caused by above-optimal temperatures (Fig. 2). Broadleaved trees displayed only modest changes in growth with increased water availability at elevated temperatures, consistent with the observed low sensitivity of the species to separate $T_{MAX}$ and CMI effects (Supplementary Fig. 3).

Soil drainage and texture significantly affected the growth response to temperature for birch, spruce species, and jack pine ($P < 0.05$; Wald test; Supplementary Table 4), and the growth response to CMI for white spruce and balsam fir ($P < 0.01$; Supplementary Table 5). Winter snowfall also conditioned how growth responded to CMI variation for all species ($P < 0.01$). Finally, stand maturity (early-seral, immature, mature, or old-growth) was also found to significantly alter climate–growth relationships for all species except white birch (Supplementary Tables 4 & 5).

**Projected responses to climate change scenarios.** We simulated species sensitivity to likely scenarios of climate change according to local site conditions. Given the uncertainty that remains in future climate conditions, we used various combinations of temperature (1–4 °C increases in $T_{MAX}$ and associated increase in PET of 43–173 mm) and precipitation changes (−5 to +15% of growing season precipitation) lying within the range of values predicted by GCMs (Supplementary Fig. 4). Future changes in temperature and associated PET as well as in precipitation were

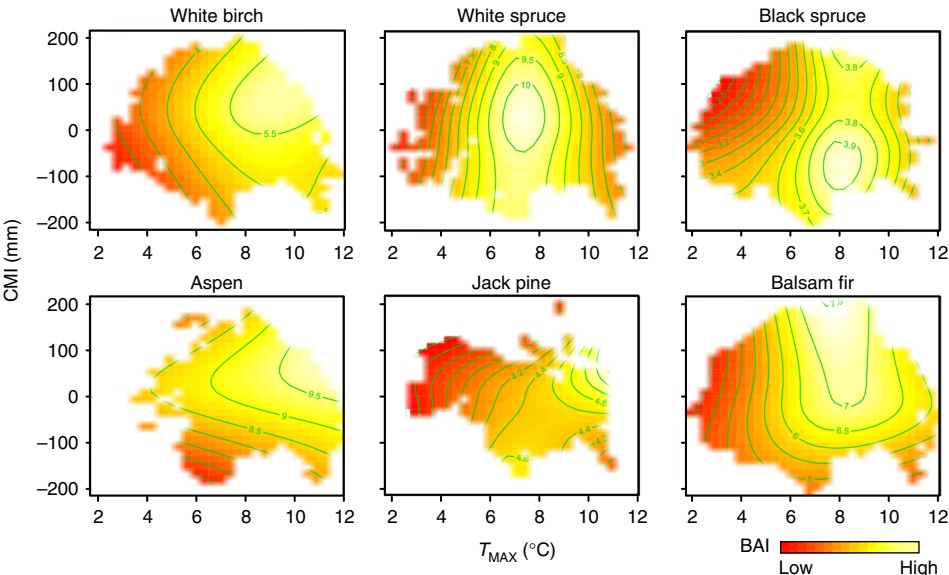

**Fig. 2** Interactive effects of temperature ($T_{MAX}$) and water availability (climate moisture index, CMI) on tree basal area increment (BAI). Heat plots indicate predicted tree BAI (in $cm^2 year^{-1}$) across observed ranges of $T_{MAX}$ and CMI, with all other model variables held at median species values

used to calculate new $T_{MAX}$ and CMI estimates that were incorporated into the growth models. Growth models were applied on a different dataset stem inventory data of over 141 million stems representative of current stand structure and composition. These simulations were limited to the colder boreal zone to avoid projecting growth responses outside the observed climate space (Supplementary Table 1). Individual growth values were summed per plot, scaled per hectare, and averaged per 15-km polygon. Results are reported relative to baseline conditions. It should be noted that climate anomalies such as drought, which may increase with climate change, are not captured in multi-decadal average changes presented here but could represent a significant component of future trends in growth and mortality.

Conifer growth displays a strong latitudinal response gradient to warming-only scenarios and its associated change in CMI (Fig. 3 and Table 2). Under a 4 °C increase in $T_{MAX}$, black spruce and balsam fir changes in growth shift from marginal declines south of 50°N to important gains north of that latitude. Similarly, jack pine growth increases across its range, but gains increase with latitude. South of 50°N, where most white spruce is found, the species growth displays large negative declines. Our projections suggest that growth decline areas expand with warming. For instance, under a 1 °C warming, black spruce growth declines are limited to southeastern Quebec and represent 3% of the species range in the boreal zone but expand to 36% with 4 °C warming (Fig. 3 and Table 2). The proportion of growth decline in white spruce shifts from 29 to 76% with a 1–4 °C change. Relative to conifers, broadleaf trees undergo modest relative changes in the southern part of the study area. North of 50°N, birch growth increases significantly with warming, while aspen growth remains largely unchanged across the entire boreal zone. Marginal birch growth declines are mainly observed in the southwest, characterized by low snowfall and precipitation, high summer temperature, and abundant glaciolacustrine deposits compared to the till-dominated deposits in the remaining territory.

Variations in future precipitation could have large impacts on boreal tree growth. For balsam fir trees south of 50°N, a 15% increase in growing season precipitation cancels out the average growth decline following a 4 °C warming (Fig. 3 and Table 2). For the same region, jack pine growth under a 4 °C warming increases from 10 ± 20% (−5% precipitation; mean ± SE; N = 548 polygons) to 31 ± 20% (+15% precipitation). Interestingly, our

projections suggest that under moderate warming, reduced precipitation could be beneficial to the growth of black spruce, white spruce, and jack pine in some high-latitude areas.

**Net growth changes in the boreal zone.** We assessed the overall change in growth to account for differences in structure, composition, and productivity across the study area. To account for the likely uneven sampling effort of the inventory data over the boreal zone (more plots in the southern boreal zone), species-specific mean growth per 15-km polygon was averaged across all polygons for each climate scenario (Fig. 4a). Aspen growth appears equally insensitive to precipitation and temperature changes, with modest net growth variations under all scenarios. Under all precipitation scenarios, white birch, jack pine, balsam fir, and black spruce display growth increases up to 2 °C warming (up to 8–13 ± 3%, N = 832–1846 polygons) albeit gains maintain or decline under additional warming depending on precipitation levels (Fig. 4a). The negative effect of increased precipitation on high-latitude jack pine and black spruce growth at moderate warming is visible at +1 and +2 °C, while additional warming inverts this trend. Precipitation levels control the growth patterns with additional warming: under a 4 °C warming scenario, jack pine growth shifts from gains of 12 ± 9% (N = 832 polygons) under 5% reduced precipitation to gains of 29 ± 10% with 15% increases in precipitation (Fig. 4a). Similarly, balsam fir and white birch net growth change is highly precipitation-dependent, with growth changes at +4 °C going from 4–6 ± 5% (−5% precipitation; N = 1307–1578) to 14–17 ± 6% (+15% precipitation). Finally, the net growth change in white spruce growth is negative under all climate simulations above 2 °C warming (with declines reaching 22 ± 7% at +4 °C; N = 959).

Plot-level growth was calculated for all species combined, averaged per 15-km polygon and over the boreal zone. The results are largely influenced by dominant species like black spruce and balsam fir, which constitute two-thirds of the inventory trees (Fig. 4b). Under 5–15% increased precipitation, a 2 °C warming results in growth gains of up to 13 ± 3% (N = 1854 polygons), while additional warming results in exponentially negative growth trends (Fig. 4b). Changes in precipitation could mitigate or exacerbate the decline, with net growth gains under a +4 °C warming doubling from 6 ± 3 to 11 ± 3% (N = 1854) under 15% increases in precipitation.

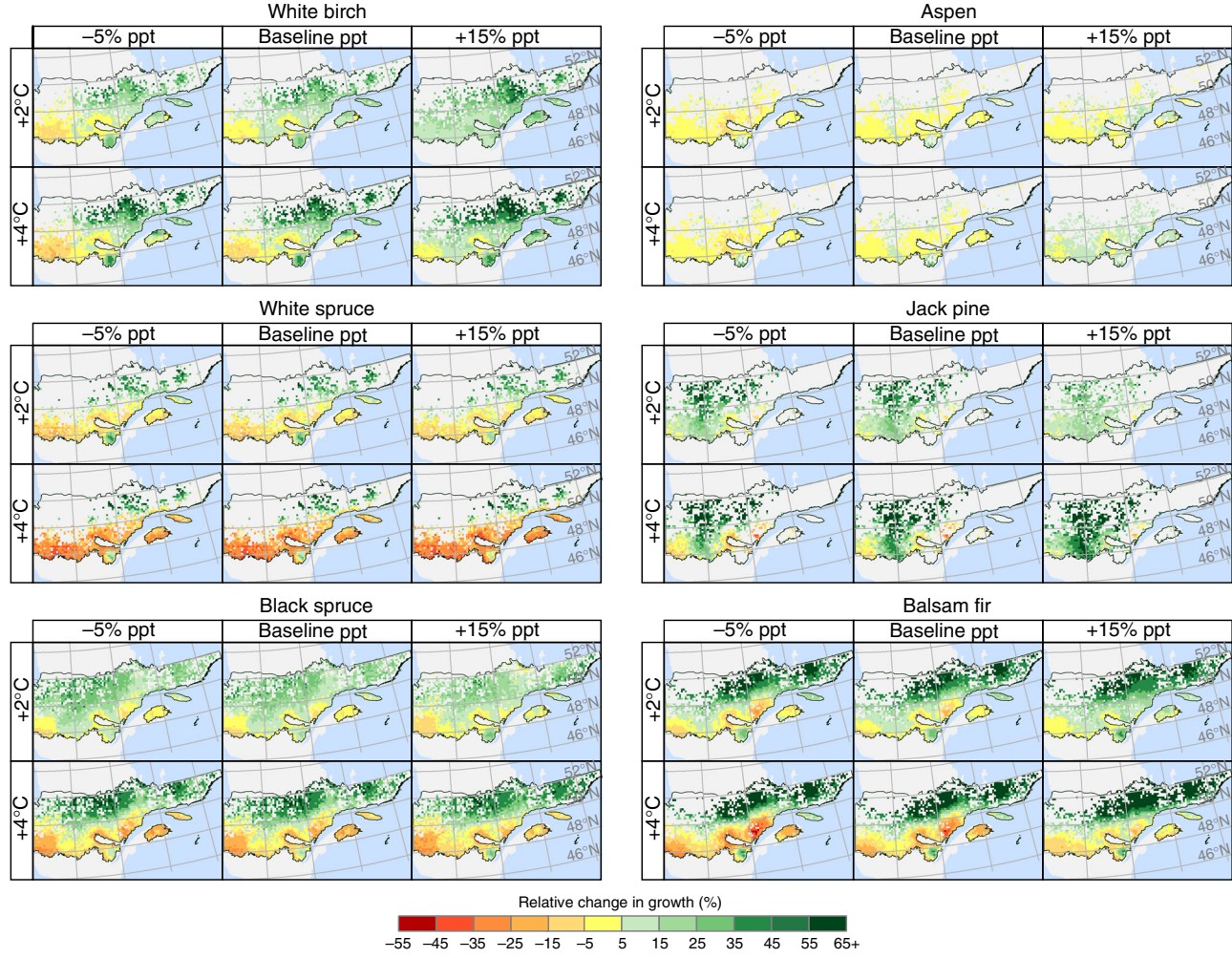

**Fig. 3** Changes in growth across Quebec's boreal vegetation zone under future climate scenarios. Relative changes in basal area growth per hectare are calculated under scenarios of 2 and 4 °C increases in $T_{MAX}$ and −5 to +15% average changes in precipitation (ppt) according to local conditions (tree size, species, mean stand age, competition, soil, slope, stand successional stage, climate). Increases in $T_{MAX}$ are accompanied by corresponding increases in potential evapotranspiration (see Methods). Values were obtained by averaging plot-level growth modeled from stem inventory data across 15-km polygons (see Methods). Data for base maps from https://www12.statcan.gc.ca/census-recensement/2011/geo/bound-limit/bound-limit-2011-eng.cfm with permission under http://open.canada.ca/en/open-government-licence-Canada and from https://www.donneesquebec.ca/recherche/fr/dataset/systeme-hierarchique-de-classification-ecologique-du-territoire used with permission under a Creative Commons 4.0—Attribution CC BY

## Discussion

The growth models developed here yield strong variance explanation and reveal the existence of climatic optimum for most studied species. Similar climatic optima were reported for black spruce from provenance trials[24,25] and are consistent with spatially-divergent responses to warming in spruce species from western North America[26]. Thermal thresholds can be explained by various physiological constraints including leaf-level water loss (see ref. [27] and references therein) and metabolic costs[28]. Indeed, we find that thermal thresholds vary significantly with water availability, highlighting the importance of analyzing both factors jointly. For most species studied here, reductions in available water increase vulnerability to elevated temperatures, as temperature is an important driver of atmospheric evaporative demand. In contrast, cold sites are more responsive to temperature and display gains in growth with lower available growing season water, consistent with findings in the Rocky Mountains of western North America[20] and in the Eurasian boreal forest[29]. The short growing season length on colder sites combined with a high quantity of snowpack meltwater may insure abundant water

levels throughout most of the growing season. At such sites, excessive precipitation combined with low PET has been reported to decrease photosynthetic activity through indirect negative effects on solar radiation, temperature, and the length of the growing season[30]. Such contrasting relationships were previously reported across the study area for black spruce populations using a classic landscape-scale dendrochronological approach[9].

The positive effect of moderate increases in temperature (1–2 °C) on boreal tree growth is consistent with the well-known temperature constraint of such forests[31]. Warming extends the growing season and increases growth rates while reducing potential cold-temperature injuries[32]. Such growth increase, in line with reported increases in black spruce growth rates north of the study area[33], may help maintain forest productivity and ecosystem services despite expected increases in future burn rates[34]. However, our simulations indicate that additional warming of 3–4 °C may cancel out part of these gains and lead to substantial growth declines in southern boreal stands conditional on future changes in precipitation levels. Growth declines are generally associated with higher probability of mortality[19–22] and

**Table 2 Projected changes in species growth rates across the southern (<50°N) and northern (≥50°N) boreal vegetation zone for likely changes in $T_{MAX}$ (+1 to +4 °C) and precipitation (ppt; −5%, baseline, and +15%)**

| | +1 °C | | | +2 °C | | | +3 °C | | | +4 °C | | |
|---|---|---|---|---|---|---|---|---|---|---|---|---|
| | −5% ppt | Baseline ppt | +15% ppt | −5% ppt | Baseline ppt | +15% ppt | −5% ppt | Baseline ppt | +15% ppt | −5% ppt | Baseline ppt | +15% ppt |
| *Growth change (% mean ± SE)* | | | | | | | | | | | | |
| White birch | | | | | | | | | | | | |
| ≥50°N (480) | 13 ± 8 | 16 ± 6 | 19 ± 10 | 25 ± 12 | 29 ± 11 | 35 ± 13 | 34 ± 17 | 38 ± 16 | 47 ± 17 | 39 ± 21 | 42 ± 21 | 53 ± 22 |
| <50°N (827) | 1 ± 6 | 5 ± 4 | 11 ± 5 | 4 ± 9 | 8 ± 7 | 15 ± 6 | 4 ± 11 | 8 ± 10 | 16 ± 8 | 3 ± 13 | 5 ± 12 | 13 ± 10 |
| White spruce | | | | | | | | | | | | |
| ≥50°N (244) | 21 ± 9 | 20 ± 8 | 14 ± 6 | 31 ± 16 | 31 ± 16 | 29 ± 14 | 33 ± 24 | 33 ± 24 | 33 ± 23 | 26 ± 29 | 26 ± 29 | 26 ± 30 |
| <50°N (715) | 1 ± 7 | 2 ± 6 | 2 ± 5 | −3 ± 10 | −3 ± 10 | −3 ± 9 | −12 ± 12 | −12 ± 12 | −13 ± 11 | −22 ± 12 | −23 ± 12 | −25 ± 11 |
| Black spruce | | | | | | | | | | | | |
| ≥50°N (947) | 15 ± 6 | 12 ± 4 | 8 ± 7 | 24 ± 9 | 23 ± 8 | 20 ± 9 | 31 ± 13 | 31 ± 12 | 29 ± 12 | 33 ± 19 | 33 ± 18 | 33 ± 17 |
| <50°N (899) | 8 ± 6 | 6 ± 4 | 4 ± 5 | 8 ± 9 | 7 ± 8 | 5 ± 8 | 2 ± 12 | 2 ± 11 | 1 ± 11 | −7 ± 13 | −6 ± 12 | −7 ± 12 |
| Aspen | | | | | | | | | | | | |
| ≥50°N (182) | 2 ± 4 | 3 ± 3 | 0 ± 4 | 3 ± 5 | 5 ± 5 | 5 ± 3 | 3 ± 6 | 6 ± 6 | 8 ± 3 | 3 ± 6 | 6 ± 6 | 11 ± 4 |
| <50°N (686) | −1 ± 4 | 1 ± 2 | 2 ± 5 | −1 ± 4 | 2 ± 3 | 5 ± 3 | −1 ± 5 | 2 ± 4 | 6 ± 3 | 0 ± 5 | 2 ± 5 | 7 ± 3 |
| Jack pine | | | | | | | | | | | | |
| ≥50°N (284) | 32 ± 13 | 27 ± 10 | 6 ± 6 | 49 ± 20 | 44 ± 18 | 26 ± 12 | 61 ± 28 | 58 ± 26 | 47 ± 19 | 68 ± 34 | 68 ± 32 | 67 ± 28 |
| <50°N (548) | 11 ± 7 | 9 ± 6 | 2 ± 4 | 14 ± 12 | 15 ± 10 | 14 ± 6 | 14 ± 16 | 17 ± 15 | 24 ± 13 | 10 ± 20 | 16 ± 20 | 31 ± 20 |
| Balsam fir | | | | | | | | | | | | |
| ≥50°N (721) | 30 ± 14 | 29 ± 12 | 25 ± 8 | 49 ± 25 | 49 ± 23 | 49 ± 18 | 59 ± 35 | 61 ± 34 | 66 ± 30 | 60 ± 43 | 63 ± 43 | 73 ± 41 |
| <50°N (857) | 4 ± 8 | 6 ± 6 | 9 ± 5 | 4 ± 12 | 7 ± 10 | 10 ± 8 | −2 ± 14 | 3 ± 13 | 7 ± 11 | −10 ± 16 | −5 ± 15 | 0 ± 11 |
| *Fraction of area with growth decline (%)* | | | | | | | | | | | | |
| White birch | 30 | 7 | 2 | 22 | 9 | 0 | 25 | 15 | 0 | 29 | 24 | 6 |
| White spruce | 34 | 29 | 27 | 50 | 49 | 49 | 67 | 67 | 68 | 75 | 76 | 77 |
| Black spruce | 3 | 3 | 18 | 11 | 11 | 14 | 22 | 22 | 24 | 37 | 36 | 37 |
| Aspen | 62 | 30 | 31 | 58 | 29 | 3 | 57 | 27 | 0 | 52 | 24 | 0 |
| Jack pine | 2 | 1 | 21 | 5 | 3 | 1 | 13 | 7 | 1 | 23 | 15 | 3 |
| Balsam fir | 15 | 6 | 1 | 22 | 15 | 6 | 32 | 24 | 16 | 43 | 37 | 29 |

The proportion of each species range displaying negative changes in growth is also summarized. Values in parenthesis indicate the number of polygons where the species is present

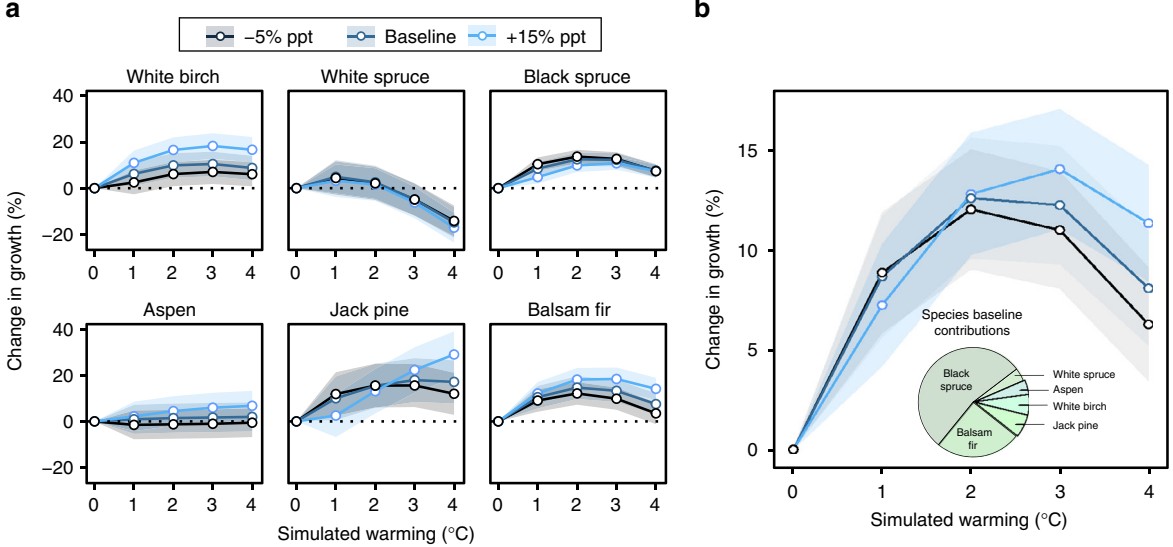

**Fig. 4** Differences in mean growth under future climate scenarios across Quebec's boreal vegetation zone. **a** Difference in mean growth per hectare per species according to 1–4 °C warming and −5 to +15% changes in growing season precipitation (ppt). Colored ribbons represent relative standard error of the mean. **b** Difference in mean growth per hectare for the combined species. Pie chart indicates the relative contribution of each species to baseline mean growth across the boreal zone. Values were obtained by averaging plot-level basal area growth across 15-km polygons, then averaging polygon-level growth across the boreal zone

could have large impacts on ecosystem dynamics, including shifts in composition towards broadleaf-dominated stands and conversion of closed-crown forests into open woodlands[16].

We report important intra- and inter-species differences in functional responses to climate. The strong sensitivity of balsam fir to drier conditions is coherent with its higher abundance in regions of high water availability, and its sensitivity to experimental drought[35]. White spruce's vulnerability to warming rather than precipitation is consistent with its dominance in the drier boreal forests of central and western Canada. Alaskan white spruce decline has been attributed to recent increases in temperature[4,36] but in Central Canada, its severe decline is rather attributed to water deficit[5]. Jack pine, balsam fir, white birch, and black spruce maintain their growth in the boreal zone under a 2 °C increase, but additional warming to 3–4 °C leads to a decline (except for jack pine under increased precipitation), while aspen

displays neutral changes. Similar results were found across a 46° to 54°N gradient in western Quebec, with positive response to warming in jack pine and black spruce in northern stands, while aspen showed neutral changes across latitudes[4]. Aspen's low sensitivity suggests that high moisture levels in Eastern Canada could preclude drought-induced declines similar to those reported in multiple locations with lower available soil moisture across North America[37].

Projecting growth responses to future conditions that fall outside the observed range of climate variability can be speculative[38,39] but we tackle this issue by predicting growth response under a range of temperature increases that lies within the observed climate space (Supplementary Table 1). Although predicting future growth trends remains limited by the uncertainty surrounding future water balance[40], this issue was considered by simulating a range of precipitation changes (−5 to +15%) based on a suite of 21 GCM simulations. However, our space-for-time modeling approach assumes that species responses to climatic gradients across space reflect their future local response to climate change over time. Our current knowledge of boreal population-level variations in climate change response remains limited[41], despite significant advances from provenance trials[42,43] or genecology studies[44,45]. In addition, tree-level growth changes are not equivalent to stand-level changes, due to complex demographics and stand dynamics, while other disturbances agents like insect outbreaks and fire will probably have large impacts on the species growth and can interact with the direct effect of drier and warmer conditions. Finally, our growth predictions do not account for likely shifts in species composition and stand structure, but our objective was rather to provide an estimate of the vulnerability of stand types that currently dominate the boreal forest. Our results point to significant regional growth declines in Northeastern North America with warming above 2 °C. Given the increasing likelihood that global warming may exceed 2 °C by the end of the century[2] and considering that it would translate into higher temperature increases at high latitude in the northern hemisphere, the capacity of boreal forests in ENA to adapt to future climate change is highly uncertain and hinges on hypothetical increases in precipitation.

## Methods

**Study area**. The study area covers the northern temperate and boreal vegetation zones of the province of Québec (Canada), which range between the 45th and the 53rd parallels north, and from the 57th to the 80th meridian west. Climate ranges from humid continental in the south, with hot and humid summers and long cold winters, to subarctic in the north, with cooler summers and longer, colder winters. Over the entire study area, mean annual temperature and precipitation for 1971–2000 range between 6.7 and −4.7 °C and between 700 and 1600 mm, respectively, while the snow-free season varies between 150 and 240 days. The study area encompasses the temperate vegetation zone in the south, largely dominated by sugar maple (*Acer saccharum*) and other broadleaf species but composed of mixed stands of balsam fir and yellow birch (*Betula alleghaniensis*) in northernmost stands. North of the temperate zone is the boreal vegetation zone, dominated primarily by black spruce and balsam fir, accompanied by white spruce, jack pine, aspen, larch, and paper birch. In addition to wood harvesting, fire and spruce budworm (*Choristoneura fumiferana*) outbreaks are the main large-scale disturbances regulating forest dynamics in these forests.

**Data collection**. The data used in this study were collected from both temporary and permanent forest plots sampled by the Québec government to characterize the managed forest territory. Forest stands were first stratified based on stand characteristics (composition, density, height, age), edaphic properties (slope, drainage, deposit), and history of disturbance from the interpretation of aerial pictures. Circular plots (radius = 11.28 m, area = 400 m$^2$) were then proportionally allocated in each stratum according to their respective surface area. Within each plot, the diameter at breast height (DBH) of all trees larger than 9 cm was measured. In addition, the DBH of all tree stems ranging between 1 and 9 cm DBH was measured within smaller circular plots (radius = 3.57 m, area = 40 m$^2$), while the DBH of all trees of DBH > 31 cm was measured within larger plots (radius = 14.1 m, area = 625 m$^2$).

Tree cores were harvested from three to nine trees of DBH larger than 9 cm, which were selected according to a strict sampling protocol[46,47]. Soil texture, deposit type, drainage, and slope were characterized during sampling. Complete core sampling and measurements, mainly conducted for site index estimation, were limited to coniferous species and to the most abundant shade-intolerant broadleaf species that generally form even-aged stands (white birch and *Populus* species). All tree cores analyzed here were collected between 1994 and 2012. To minimize growth bias caused by disturbances like spruce budworm outbreaks (1970–1987 and 2006–today), growth was limited to years 1985–2005.

**Tree-ring data preparation**. Cores were dried, glued to a wooden holder, and sanded according to standard procedures[48]. Ring boundaries were first detected and identified under binocular magnification and then measured to the nearest 0.01 mm with the WinDendro Image Analysis System for tree-ring measurement (Regent Instruments Inc.). A calendar year was attributed to each ring, the outermost ring corresponding to the year of tree sampling, or exceptionally to the year before for plots sampled prior to the start of tree-ring formation. For each tree, Tukey's test was used to detect outliers (with constant $k = 3$) based on the distribution of annual growth values[49]. Abnormal annual growth values, likely caused by anomalous ring detection, represented 0.4% of growth-year values and were generally evenly distributed across years. All abnormal values were excluded from analysis.

Radial growth of 270,615 trees, covering 95,562 plots and representing the seven typical boreal tree species of Eastern Canada (black and white spruce, balsam fir, jack pine, paper birch, aspen, and larch), was converted into annual basal area increment (BAI), in cm$^2$ year$^{-1}$, from the bark to the pith using tree DBH as the initial value. Annual BAI of each tree was averaged over the 15 most recent years of growth within the 1985–2005 period. Trees with 10–14 years of growth were also included (e.g., a tree sampled in 1997, for which growth prior to 1985 was excluded, would only display 13 years of growth). We assumed that a shorter period (<10 years) would give too much weight to anomalous growth years, while a longer period (>15 years) would increase the risk that competition, measured during the year of stand sampling, would no longer be representative of earlier growth conditions.

The most represented species in our tree-ring collections are black spruce (91,811 trees) and balsam fir (89,097), followed by paper birch (37,526), aspen (21,905), white spruce (15,262), jack pine (12,068), and larch (1946; Table 1). Consistent with their larger size, white spruce (9.2 cm$^2$ year$^{-1}$) and aspen (8.3 cm$^2$ year$^{-1}$) have the highest median growth, followed by larch (6.5 cm$^2$ year$^{-1}$) and balsam fir (5.8 cm$^2$ year$^{-1}$), while black spruce (2.9 cm$^2$ year$^{-1}$) displays the lowest growth rates (Table 1 and Supplementary Fig. 2). Half of the sampled trees (50%) are found in the boreal forest zone while the remaining half is in the northern temperate forest zone (Fig. 1). Annual tree diameter, referred to hereafter as tree size, was estimated by subtracting annual diameter increment from initial, measured DBH. Annual tree diameter was then averaged over each tree's 15 most recent years of growth within the 1985–2005 period.

Tree age was estimated as the sum of observed tree rings. Potential biases in tree age due to core decay were avoided as incomplete core samples were discarded. Most larch trees are less than 50 years old (median = 35, Table 1), while other species have similar age structures between 42 and 54 of median age, except for black spruce (median: 67 years, Table 1 and Supplementary Fig. 2). Black spruce displays a higher proportion of older trees, with a 95th percentile age of 171, relative to 82–121 for other species (Table 1 and Supplementary Fig. 2). This is mostly due to the stands sampled north of 50°N, which are dominated by older black spruce trees (median age of 132 years). Tree age was averaged over each tree's 15 most recent years of growth within the 1985–2005 period.

**Climate**. Monthly weather data was generated for each plot using the BioSIM interpolation model[50] based on a network of 249 (1985) to 365 (2005) weather stations. To assess the control exerted by available water on growth, monthly PET was estimated with the SPEI package in R[51] using the Penman–Monteith algorithm with inputs of monthly average daily minimum and maximum temperature, latitude, incoming solar radiation, temperature at dew point, and altitude. Relative to other evapotranspiration algorithms, the Penman–Monteith algorithm is a more accurate, comprehensive, and physically based model of PET[40]. The CMI was calculated as the balance of PET and precipitation over a period $i$, in mm of water (Eq. (1)):

$$CMI_i = Prec_i - PET_i \qquad (1)$$

The CMI is a hydrologic index well-correlated with tree growth in boreal and temperate forest ecosystems[52,53].

Average summer (CMI$_{SUMMER}$; June–August) and growing season (CMI$_{GS}$; May–September) climate moisture indices were used to assess the documented warm-season water constraint on growth[4–7], while average mean ($T_{MEAN}$) and maximum ($T_{MAX}$) daily temperature estimates were used to assess the control exerted by annual temperature on growth[15]. Across the study area, CMI$_{GS}$ is poorly related to $T_{MAX}$ ($R^2 = 0.06$). All climate variables were averaged over the 11–15 years of growth for each tree. Since species-specific sensitivities to interactions of

temperature and water are the focus of this study, we used only one variable to model temperature and one to model water availability. Variable selection is described in the Model section below. Along the growing season CMI gradient (range: −205 to 219 mm), the gradient common to all species extends from −125 to 161 mm (Supplementary Fig. 2). All species samples display similar distributions along CMI values, although balsam fir samples are more represented than other species at high CMI values (95th percentile = 145 versus 60–116 mm for other species; Table 1). All species samples are found to span $T_{MAX}$ values between 3.6 and 11.1 °C (Supplementary Fig. 2), while the $T_{MAX}$ gradient over the study area ranges between 1.8 and 11.9 °C (Fig. 1). Black spruce and jack pine samples are also less abundant at high $T_{MAX}$ (95th percentiles of 8.7–9.2 °C) than other species (95th percentiles of 9.5–10.6 °C; Supplementary Fig. 2 and Table 1). Along this gradient, black spruce, jack pine, and balsam fir are more abundant at low $T_{MAX}$ (5th percentiles of 3.5–4.7 °C) than other species (5th percentiles of 5.6–6.3 °C; Supplementary Fig. 2 and Table 1).

Because spring snowmelt can have lasting effects on the growing season available water, the amount of snowfall—and thus the amount of snowmelt—was estimated as the total snowfall from January to March. Median January to March snowfall values range from 144 to 180 mm (Table 1). White birch, white spruce, and balsam fir samples are less represented at sites with low snowfall (5th percentile > 128 mm versus 112–114 mm for other species), while jack pine is less represented at sites with high snowfall (95th percentile = 166 mm versus 208–251 mm for other species).

**Soil physical environment.** Local slope, drainage, texture, and soil deposit type were determined in situ to characterize the soil physical environment[46]. Missing values were completed using the interpretation of aerial photographs. Slope was estimated quantitatively, as a percent, in all plots. However, slope values in temporary sample plots were converted into classes of 0–3, 4–8, 9–16, 16–30, 31–40, and ≥ 41%. For this analysis, we converted these classes to quantitative values using the lower range value (e.g., 9–16 was replaced with 9%). A qualitative combination of drainage, texture, and soil deposit type was used during plot sampling to estimate the soil physical environment. Sites on very thin (<25 cm) or very stony (>80% stoniness) soils were defined as shallow or stony soils. Sites characterized on site by organic soils or mineral soils with hydric moisture regime were defined as hydric soils. The remaining mineral soils were divided into six classes depending on their drainage (xeric to mesic or hygric) and soil texture (coarse, medium, or fine), all determined on site using standardized protocols. Hygric soils display permanent seepage and mottling and some gleyed mottles in the soil profile, while xeric to mesic soils have a more rapid drainage.

Larch and jack pine excepted, all sampled trees are most frequent on well-drained, medium-textured soils (39–69% of trees; Supplementary Fig. 2). Hydric soils are the most common environment for larch (51% of sampled trees), followed by black spruce (20%), but only marginal for the remaining species (1–5%). Larch presence is also associated with flat landscapes (median slope of 0) relative to other species (Table 1). A high proportion of jack pine (42% of trees) is found on mesic, coarse-textured soils (e.g., sandy soils) as compared to other species (4–12%).

**Competition.** To account for competition for resources from neighboring trees, two competition indices were estimated for every sampled tree. First, symmetric competition (BA) was computed as the sum of all individual basal areas for trees with DBH > 1 cm, scaled to a hectare (units of $m^2 ha^{-1}$). To account for the size of the sampled trees relative to the size of competing trees—assuming the level of competition exerted by a smaller tree is lower—asymmetric competition (BAL) was computed by summing only the basal area of trees that were larger than the cored tree[54]. Competition levels and their distribution are similar across species, with median BA values ranging between 24 (larch and jack pine) and 30 $m^2 ha^{-1}$ (balsam fir), and median BAL values ranging between 9 (black spruce and jack pine) and 17 $m^2 ha^{-1}$ (balsam fir; Table 1 and Supplementary Fig. 2).

**Growth model.** GAMs were used to estimate the joint effects of temperature and water availability on growth and to detect non-linear climatic relationships. These models are semi-parametric extensions of generalized linear models which fit adjusted non-linear smoothing terms using regression splines as predictors without any a priori assumption on the relationship[55]. Thus, GAMs are particularly useful to detect non-linear responses and thresholds, and to predict species response to climate across its range[56]. A single GAM was fitted for each species to predict the annual BAI of a tree $j$ at a site $i$ as a function of temperature, water availability, competition, tree size and age, snowfall, and soil physical environment, assuming a Gaussian distribution of the response variable (Eq. 2):

$$\log\left(BAI_{ij} + 1\right) = \beta + \log\left(Size_{ij}\right) + Size_{ij}$$
$$+ f(Age_{ij}) + f(BAL_{ij}) + f(BA_i)$$
$$+ f(Temperature_i) + f(CMI_i) + f(Temperature_i, CMI_i) \quad (2)$$
$$+ f(Slope_i) + \epsilon_{soil} + \epsilon_{snow} + \epsilon_{stage}$$

where $\beta$ is the intercept and $f$ are smoothing functions represented by cubic regression splines. To minimize over-fitting and the complexity of the model, the

degree of smoothness of the spline functions was bounded to four for each variable[56]. The interaction between $T_{MAX}$ and CMI was modeled with a tensor product smooth independent of the relative scaling of covariates with a degree of smoothness bounded to three[57].

BAI was log-transformed to avoid negative growth predictions and normalize model residuals. Back-transformed BAI estimates are presented therein, after applying the Smearing retransformation to correct the potential bias associated with the transformation of a predicted variable prior to estimation[58]. Following visual inspection of univariate relationships, tree size was included as a 2nd degree polynomial[59]. More complex relationships were observed for other factors, which were thus modeled using smooth functions. Tree age was included in the growth model to correct for the well-documented sampling bias caused by the absence of old, fast-growing trees and young, slow-growing trees from the tree ring dataset (see Supplementary Note 1). Due to the high spatial autocorrelation for abiotic factors, we did not include an explicit spatial structure in the model.

Based on the low spatial autocorrelation of the model residuals (Supplementary Fig. 5), no explicit spatial structure was included in the model. Three error terms were included in the model. Soil characteristics (combined soil drainage and texture) were included as a seven-level error term associated with the origin of the model and the smooth functions $f$ for $T_{MAX}$ and CMI. Average January to March snowfall was converted into a three-level error term associated with the origin of the model and the CMI function, with low snowfall below 140 mm (21% of sites), medium snowfall between 140 and 200 mm (57% of sites), and high snowfall above 200 mm (22% of sites). Finally, stand successional stage was included as a four-level error term derived from the average tree age for each stand (early-seral: 0–20 years, immature: 20–70 years, mature: 70–100 years, old-growth: >100 years) and associated with the origin of the model and $T_{MAX}$, CMI, BA, and BAL smooth functions. While tree responses to climate have been reported to change with stand successional stage[60], we also observed during preliminary analysis a changing influence of competition on tree growth with stand successional stage, especially for black spruce and jack pine. Most stands (41–75% across species) fall in the immature stage, 17–29% in the mature stage, early-seral are the least abundant (1–6%), while old-growth are generally scarce (2–12%), except for black spruce, which is well represented in old stands (34%; Supplementary Fig. 2). We observe an initial short-lived increase in stem density for all species except white spruce (decline) and jack pine (no changes), followed by a decline with stand development stage in all species, consistent with self-thinning theory (Supplementary Table 2).

**Climatic variable selection and model validation.** To select the best descriptive climatic variables, Akaike information criterion (AIC) values[61] were compared between models with different combinations of temperature ($T_{MAX}$ and $T_{MEAN}$) and water availability variables ($CMI_{SUMMER}$ and $CMI_{GS}$). Retained variables were average daily maximum temperature ($T_{MAX}$, in °C) and growing season CMI (May–September, in mm; Supplementary Table 3). The two variables display contrasted spatial structures over the study area, with temperature following a latitudinal gradient and growing season CMI varying longitudinally (Fig. 1). All other factors in the final model (e.g., competition, tree age) were included based on the authors' ecological understanding of the study area. All variables reduced the AIC value, indicating an improved model despite added complexity (Supplementary Table 3). For each species, the initial model was fit on a random subset of 80% of the trees. The predictive capacity of the model was then validated using the remaining 20% of the trees, computing the explained deviance and the root mean square error (RMSE) of predicted versus observed growth rates. The model was then fit to all trees of each species.

**Simulated growth change with warming and drying.** Future changes in temperature and precipitation associated with global warming were calculated from 21 GCMs from the NASA Earth Exchange Global Daily Downscaled Projections (NEX-GDDP) dataset (resolution is 0.25°). Low and high scenarios of future greenhouse gas emissions (Representative Concentration Pathways, RCP) of 4.5 and 8.5 W m$^{-2}$ were used. Given the uncertainty that remains in future climate conditions, species growth models were then used to compare species sensitivity to various combinations of temperature (1–4 °C increases in $T_{MAX}$ and associated increase in PET of 43–173 mm) and precipitation changes (−5 to +15% of growing season precipitation) within the range of predicted values by the GCMs (Supplementary Fig. 4). Future changes in temperature and associated PET as well as in precipitation were used to calculate new $T_{MAX}$ and CMI estimates that were incorporated into the growth models.

Each combination of temperature and precipitation change was calculated as the average over the boreal study area, but these changes were allowed to vary locally according to median GCMs projections. By doing so, our simulations account for the projected heterogeneous changes in climate projected by GCMs. Notably, projections suggest the rate of warming over the boreal study area is faster at high latitudes, while eastern parts of the study area, closer to the Atlantic Ocean, will receive most of the potential increases in precipitation (Supplementary Fig. 4). Growth models were applied to a different dataset that is representative of current stand structure and composition. Within all forest inventory plots where tree-ring collections were sampled, all stems >9 cm DBH were classified into 2-cm DBH classes per species, for a total of more than 141 million stems of the study species. Levels of BAL were computed for each stem. We then simulated the growth of each

stem of each species under study according to local conditions (competition, soil, climate, tree size, and stand-level mean age), after adding local changes in $T_{MAX}$ corresponding to the average 1–4 °C warming across the study area, as well as adjusting local CMI using corresponding changes in PET associated with temperature increases, combined with local changes in May to September precipitation corresponding to average changes of −5, +5, and +15%.

Resulting individual growth estimates were converted into growth per hectare, in cm$^2$, and summed per plot. To account for the likely uneven sampling effort of the inventory data over the boreal zone (more plots in the southern boreal zone), a 15-km grid was applied on the boreal zone, and growth was averaged per polygon. To account for structural, composition, and productivity differences across the boreal zone, the overall change in growth was computed by averaging polygon-level growth across all polygons for each climate scenario. All results were reported relative to baseline conditions.

Anticipating forest response to climate change can be highly problematic when predictions are made for climatic conditions beyond the data used to fit a model[38,39]. Here, projected responses to temperature increases of 1–3 °C remain within the observed range of $T_{MAX}$ across the entire boreal study area for all species (Supplementary Table 1). For a 4 °C increase in $T_{MAX}$, the fraction remains above 97% for all species except jack pine (>85%). A 4 °C increase in $T_{MAX}$ in these stands is 0.2 ± 0.2 °C (mean ± SD) above observed temperatures. Additional analyses were performed excluding these areas from future growth projections, with marginal effects on the general trends reported here, suggesting that our projections for the boreal zone are robust.

**Code availability**. The code used to fit the growth models and project future growth trends can be made available upon request.

**Data availability**. The environmental data that support the findings of this study are available from the Ministère des Forêts, de la Faune et des Parcs du Québec (MFFPQ) at https://mffp.gouv.qc.ca/le-ministere/acces-aux-donnees-gratuites/. The tree-ring data that support the findings of this study were used under license for the current study but are however available from the authors upon reasonable request and approval by the MFFP. Finally, the climate scenarios used are available at https://cds.nccs.nasa.gov/nex-gddp/.

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

## Acknowledgements

Funding for this study was provided by a postdoctoral scholarship to L.D'O. by the MITACS Accelerate program and by the MFFP and Le Fond Vert du Ministère du Développement Durable, de l'Environnement et de la Lutte contre les Changements Climatiques du Québec within the framework of the Action Plan 2013–2018 on climate change. We gratefully acknowledge the staff of the Ministère des Forêts, de la Faune et des Parcs du Québec (MFFP) for the fastidious work related to tree core sampling, preparation, and measurements, Marie-Claude Lambert who generated meteorological data and L.E. Robert for modeling advice. We thank Travis Logan at the Ouranos Consortium on Regional Climatology and Adaptation to Climate Change for processing climate scenarios from the NEX-GDDP dataset, prepared by the Climate Analytics Group and NASA Ames Research Center using the NASA Earth Exchange and distributed by the NASA Center for Climate Simulation.

## Author contributions

L.D'O., L.D., and D.H. designed the study and methodology with substantial inputs from D.K. L.D'O. obtained and analyzed the data. L.D'O. wrote the first draft with substantial inputs from D.H. L.D'O., D.H., L.D., R.P.P., Y.B., and D.K. contributed to data interpretation and manuscript preparation.

## Additional information

**Competing interests:** The authors declare no competing interests.

