## [Peer Review File · Nature Communications]

Reviewers' comments:

Reviewer #1 (Remarks to the Author):

This is an important study that is based on a broad dataset and a sound statistical approach leading to interesting results that contribute to the body of knowledge. My comments are mostly related to the climate scenarios and how climate warming is represented in the study on a spatial scale. One of the main results of the study is depicted in fig.3, where the changes of the growth across the boreal zone is shown. The figure represents different variations of tmax increases and changes in precipitation. On p.16 the authors state that have simulated climate change using 21 downscaled and averaged GCMs for two RCPs - the distribution of the results is shown in figure S4 - I am wondering what the connection between these simulations and fig.3 is - fig.3 seems to depict an equal warming over the whole study area, which would not be in line with the downscaled GCMs - this should be clarified. On p.17, the authors state that climate models do not project homogeneous changes as assumed in this study, I am now wondering what the purpose of the climate projections in this study is.

On p. n16 (l.432) the authors claim that consistent with self-thinning theory, stem density steadily declines with stand development stage. Looking at table S1 this is only partly true: for white birch, black spruce and Balsam fir the density increases with some stages, some of the increases are even significant, for Balsam fir the old-growth stage has even higher stem densities than the early -seral stage - overall, I find the stem-densities extremely high and I am wondering where this comes from and what the diameter threshold is in this table and how this increase can be explained.

On p.16 the authors state that they did not include an explicit spatial structure in the model due to the high spatial autocorrelation for abiotic factors - I find this rather obscure as an explanation - if there is spatial autocorrelation, the model should take that into account, this should be clarified.

I have a few minor comments that rather deal with formal issues:

table 1 (especially the caption) is not really straightforward - the explanation - "values in parenthesis (validation)..." is misleading - most of the values in parenthesis refer to 5th,9th percentiles. In fig.1a most of the green plots are in the temperate zone, but the paper deals with the boreal zone, I don't understand why they are plotted..

fig. 2 what is the unit for BAI (low-high ... is too imprecise), what are the isolines in the figure depicting?

I would expect the scientific (Latin) names of the involved species to appear the first time they occur in the text and not only in the method section - if this is required by the journal, I would put a hint that the scientific names come later.

p.17, l.481 - Data availability

Reviewer #2 (Remarks to the Author):

Review of Manuscript NCOMMS-18-08320-T

Overview

This manuscript reports on a significant effort to model forest growth in the boreal region of Quebec, Canada as a function of climate and various site attributes – and then project future growth under several climate change scenarios. The manuscript is well written, well organized, and reports on an important topic in climate change science. While other studies have projected forest growth under climate change, this effort appears unique in the degree to which it explores the interplay between temperature and moisture in shaping the growth response of boreal trees. Below I request clarification and expansion on a few topics and provide a number of minor comments for the authors' consideration.

Main Comments

1. My understanding is that, with the GAM approach used here, extrapolations cannot be made outside the range of the data used to develop the model. The authors demonstrate in Table S3 that, under the various climate change scenarios, the majority of the boreal portion of the study area (i.e., the region for which the growth projections are being made) is still within the temperature domain of the data used to develop the model. However, under the +4°C scenario, some species have data that fall outside that temperature domain (e.g., 22.6% of jack pine observations, Table S3). It is not clear to me how the GAM model was used to make these extrapolations – and whether it is appropriate, even in this limited usage.
2. It is not entirely clear to me why the authors used CMI as opposed to precipitation to represent moisture in their model. They present the growth projections under climate change (Fig. 3 and related text) in terms of changes in precipitation (i.e., -5% and +15%), presumably because this is easier for readers to relate to than changes in CMI. I'm wondering if they explored the possibility of using precipitation directly in their models; this would also have the advantage of being more independent from temperature than CMI (which is calculated using temperature variables).
3. I see only one mention in the ms of population-level variation in climate change response (Lines 236-239) – with the assumption here being that populations are going to respond consistently across the range of the species. I'm not opposed to this, but this is a key assumption that probably warrants further elaboration – at least recognizing the significant work done in this area vis-à-vis provenance trials, genecology, etc.

Specific Comments

1. The amount of deviance explained is (unnecessarily) repeated in lines 69 and 71.
2. Lines 84-87; may be worth mentioning that the linear relationships between growth and climate for certain species are surely due to a lack of samples from one end of the climatic gradient.
3. Line 137; replace 'from and' with 'from'.
4. Lines 134-144; when discussing impacts of precipitation on future tree growth it may be worth reminding readers that this study is examining responses to changes in average (multi-decadal) precipitation conditions. Extreme events (droughts and floods), which may increase with climate change and which are not captured in multi-decadal averages, could also have significant impacts on forest growth and mortality.
5. Line 235; consider removing the word 'somehow' from this sentence.

Reviewer #3 (Remarks to the Author):

In this study the authors apply growth models to the 7 dominant tree species in the eastern Canada boreal region. They found that climate sensitivity was species specific and the dominant climatic limitations varied with latitude. The authors identify climatic thresholds in order to predict future trajectories of this ecosystem in response to climate change. I feel that this work is very interesting and an important contribution to our understanding of the future of the boreal ecosystem. The manuscript was very well written and I have a few minor comments listed below. The only substantive comments I have are in regards to the models. I think the use of a general additive model is great, but my concern lies in the exclusion of random effects. In addition, I feel that clarification of the simulated growth model in regards to the inclusion of CMI is needed (see comments below).

L 33 – delete increasingly

L36 – 39. This sentence has a lot of information. Perhaps start a new sentence after “warming”.

L52 – what are the 7 species?

L74 – P and PET have not been defined above – reference the reader to the methods?

L75 – interesting that larch is insensitive to climate and competition – I wonder what is limiting larch growth? Are these results in supplementary?

L85 – remove “e.g.”

L 118 – 144 – these results were a bit hard to wade through. I know the trends are presented in the figure – but perhaps the numbers could be condensed into a table to simplify and just highlight the most important trends rather than the exact numbers in the text?

L280 – I’m curious as to why 9 cm DBH was used – based on growth rates it would seem as though a highly productive tree (white spruce) would be much younger than a tree of very low productivity (black spruce) once this 9cm threshold was reached. I assume that the inclusion of age in the models accounts for this – which is written in the equation on L 403 but not in the text preceding it. Furthermore, sampling only these trees creates a bias that you explain nicely in L715-719 but should be referenced in the text.

L417 – but you do have uneven sample sizes associated with this spatial structure, correct? If so, I think that the random effect of site should either be included in your models or should at least be tested and then if insignificant removed.

L 462 - In reading the main text, I was curious as to why CMI was used in the GAM, but precipitation was used in the simulated growth model. This sentence alludes to the inclusion of CMI in the simulated model, but I don't quite understand how – perhaps include the model formula? I think this needs to be expanded/clarified and should probably be included in the main text.

Response to the Referees

The following is a detailed description of the changes made and our response to each of the reviewers' points.

Reviewer 1

REVIEWER - This is an important study that is based on a broad dataset and a sound statistical approach leading to interesting results that contribute to the body of knowledge. My comments are mostly related to the climate scenarios and how climate warming is represented in the study on a spatial scale. One of the main results of the study is depicted in fig.3, where the changes of the growth across the boreal zone is shown. The figure represents different variations of tmax increases and changes in precipitation. On p.16 the authors state that have simulated climate change using 21 downscaled and averaged GCMs for two RCPs - the distribution of the results is shown in figure S4 - I am wondering what the connection between these simulations and fig.3 is - fig.3 seems to depict an equal warming over the whole study area, which would not be in line with the downscaled GCMs - this should be clarified. On p.17, the authors state that climate models do not project homogeneous changes as assumed in this study, I am now wondering what the purpose of the climate projections in this study is.

AUTHORS – The rationale used in our analysis was to test the species sensitivity to various fixed levels of changes in temperature and precipitation which are in line with downscaled GCM simulations for the study area. Although the original approach was adequate to assess species sensitivity to climate change across their range, we agree with the reviewer that this approach is less ideal when used to project future net changes in growth for the boreal zone (Figures 3 and 4), since regional variations in climate change are not accounted for.

Therefore, we recalculated future growth changes for the same 1-4°C increase scenarios in annual T_{MAX} and -5 to +15% in growing season precipitation. This time, the scenarios represent the average change in climate across the boreal zone but incorporate regional differences in climate (with 20x20 km polygons) as projected by downscaled GCMs. For instance, an average 4°C increase in temperature over the boreal zone represents increases ranging from 3.55 to 4.21°C across the area, while a +15% increase in precipitation represents local increases ranging between 10.9 and 21.1%.

We believe this revised approach is a significant improvement over our preceding approach. The general trends reported here remain the same, but we observe that the growth declines reported for southern stands are less severe, while the growth increases reported for northern stands are exacerbated. This is caused by projected gradients in temperature and precipitation changes over the study area: relatively lower warming in the south (less detrimental), stronger warming in the north (larger gains), and the larger increases in precipitation in eastern stands where balsam fir is more abundant, which favours its growth.

We have included a description of the methodological changes in the Methods section, as well as a map of projected gradients in temperature and precipitation changes in the Supplementary Materials (Fig S4). The text now reads:

Methods: (P19 L467): « *Given the uncertainty that remains in future climate conditions, species growth models were then used to compare species sensitivity to various combinations of temperature (1-4°C increases in T_{MAX} and associated increase in PET of 43-173 mm) and precipitation changes (-5 to +15% of growing season precipitation) within the range of predicted values by the GCMs (Figure S4). Future changes in temperature and associated PET as well as in precipitation were used to calculate new T_{MAX} and CMI estimates that were incorporated into the growth models.*

Each combination of temperature and precipitation change was calculated as the average over the boreal study area, but these changes were allowed to vary locally according to median GCMs projections. By doing so, our simulations account for the projected heterogeneous changes in climate projected by GCMs. Notably, projections suggest the rate of warming over the boreal study area is faster at high latitudes, while eastern parts of the study area, closer to the Atlantic Ocean, will receive most of the potential increases in precipitation (Figure S4). Growth models were applied to a different dataset that is representative of current stand structure and composition. Within all forest inventory plots where tree-ring collections were sampled, all stems >9cm DBH were classified into 2-cm DBH classes per species, for a total of more than 141 million stems of the study species. Levels of BAL were computed for each stem. We then simulated the growth of each stem of each species under study according to local conditions (competition, soil, climate, tree size and stand-level mean age), after adding local changes in T_{MAX} corresponding to the average 1-4°C warming across the study area, as well as adjusting local CMI using corresponding changes in PET associated with temperature increases, combined with local changes in May to September precipitation corresponding to average changes of -5, +5, and +15%. »

REVIEWER - On p. n16 (L432) the authors claim that consistent with self-thinning theory, stem density steadily declines with stand development stage. Looking at table S1 this is only partly true: for white birch, black spruce and balsam fir the density increases with some stages, some of the increases are even significant, for Balsam fir the old-growth stage has even higher stem densities than the early -seral stage - overall, I find the stem-densities extremely high and I am wondering where this comes from and what the diameter threshold is in this table and how this increase can be explained.

AUTHORS – Thanks to the reviewer, we found and corrected an error in our function to scale per hectare measured stem density. After correcting this error, stem density values are now significantly lower. Since stem density was not incorporated in our models and only used here to describe stand developmental stages, this error has no effects on our findings. Initially, stem density was calculated for all stems with DBH>1cm. We have now limited our stem density calculations to stems with DBH>9cm as this is the most common size threshold value for trees to be considered represents ‘merchantable’ stems. We observe an initial short-lived increase in stem

density for all species except white spruce (decline) and jack pine (no changes), followed by a decline in stem density in all species, consistent with self-thinning theory. We have corrected Table S1 and added information in the Methods section:

(P18 L446) « *We observe an initial short-lived increase in stem density for all species except white spruce (decline) and jack pine (no changes), followed by a decline with stand development stage in all species, consistent with self-thinning theory (Table S1).* »

REVIEWER - On p.16 the authors state that they did not include an explicit spatial structure in the model due to the high spatial autocorrelation for abiotic factors - I find this rather obscure as an explanation - if there is spatial autocorrelation, the model should take that into account, this should be clarified.

AUTHORS – Following the reviewer’s comment, we tested for the presence of spatial autocorrelation in the model residuals. If it was the case, the models would require accounting for this spatial autocorrelation. Results from this analysis, now included in the supplementary materials of the manuscript (Fig. S5), clearly demonstrate the absence of significant spatial autocorrelation in the model residuals. Therefore, we have not included spatial random structure in our models.

We have added this information in the Methods section:

(P18 L431): « *Based on the low spatial autocorrelation of the model residuals (Figure S5), no explicit spatial structure was included in the model.* »

REVIEWER - I have a few minor comments that rather deal with formal issues:

REVIEWER - table 1 (especially the caption) is not really straightforward - the explanation - "values in parenthesis (validation)..." is misleading - most of the values in parenthesis refer to 5th,9th percentiles.

AUTHORS – We agree with the reviewer and have corrected table 1 (and its caption) accordingly.

REVIEWER - In fig.1a most of the green plots are in the temperate zone, but the paper deals with the boreal zone, I don't understand why they are plotted.

AUTHORS – Figure 1a displays the location of all plots that were used to fit our growth models. Indeed, our models were fitted to growth observations extending to the temperate zone (south of the boreal zone) to avoid the common caveat of modelling outside the observed climate space (see Table S3). Thus, instead of extrapolating/projecting growth responses outside of our observed climate, we used observed species response to warmer climate from the temperate zone to reduce speculations and better estimate the response of the boreal zone to future climate scenarios. We agree with the reviewer that this should be more clearly stated in the paragraph where we refer to Figure 1a. We have made some changes to the paragraph accordingly:

(P2 L55) « *First, general additive models were used to model the growth (1985-2005) of 270,000 trees across 95,000 temperate and boreal stands (Figure 1), while accounting for local climate, tree size and age, soil characteristics, successional stage and competition with neighboring trees. Second, models were used to assess the local vulnerability of 141 million inventoried stems in the boreal zone under study to an array of temperature and precipitation change scenarios based on a suite of general circulation models (see SM). The fact that our models were fitted to growth observations extending to the warmer temperate zone allowed us to simulate boreal growth responses to warming while remaining within the observed climate space (Table S3). »*

REVIEWER - fig. 2 what is the unit for BAI (low-high ... is too imprecise), what are the isolines in the figure depicting?

AUTHORS – We have changed Figure 2 to include values with the isolines in the figure.

REVIEWER - I would expect the scientific (Latin) names of the involved species to appear the first time they occur in the text and not only in the method section - if this is required by the journal, I would put a hint that the scientific names come later.

AUTHORS – Correct. The latin names of the species were added to the manuscript when they first appeared in the text.

REVIEWER - p.17, l.481 - Data availability

AUTHORS – The title was corrected.

Reviewer 2

REVIEWER 2 - Overview:This manuscript reports on a significant effort to model forest growth in the boreal region of Quebec, Canada as a function of climate and various site attributes – and then project future growth under several climate change scenarios. The manuscript is well written, well organized, and reports on an important topic in climate change science. While other studies have projected forest growth under climate change, this effort appears unique in the degree to which it explores the interplay between temperature and moisture in shaping the growth response of boreal trees. Below I request clarification and expansion on a few topics and provide a number of minor comments for the authors' consideration.

REVIEWER 2 - 1. My understanding is that, with the GAM approach used here, extrapolations cannot be made outside the range of the data used to develop the model. The authors demonstrate in Table S3 that, under the various climate change scenarios, the majority of the boreal portion of the study area (i.e., the region for which the growth

projections are being made) is still within the temperature domain of the data used to develop the model. However, under the +4°C scenario, some species have data that fall outside that temperature domain (e.g., 22.6% of jack pine observations, Table S3). It is not clear to me how the GAM model was used to make these extrapolations – and whether it is appropriate, even in this limited usage.

AUTHORS – We agree with the reviewer that extrapolations outside the range of data used to develop the model are highly speculative for all non-process-based models, not only the GAM approach developed here.

Following the changes in future climate scenarios that were suggested by reviewer 1, the fraction of jack pine area that falls outside the observed climate domain is now reduced from 22.6% to 14.8% (Table S3 was updated accordingly). This potential issue is further reduced by the fact that the mean temperature of this area is only $0.2 \pm 0.2^\circ\text{C}$ (mean \pm SD) above observed temperatures.

Nonetheless, to address the reviewer's concern, we have recalculated values used to plot Figure 4 while excluding the areas falling outside the observed climate space. The removal of these areas does not change the general trends reported here. Since these areas are located in the warmest part of the boreal zone, removing them increases the relative weight of northern stands, where most of the growth increases are observed. For instance, projected net growth changes for jack pine under a 4°C warming shift from $12-29 \pm 10\%$ to $17-34 \pm 12\%$ without these areas. Therefore, we chose to keep these areas in our projections but have added significant additional information in the Methods section of the main text:

(P19 L495) : « *Anticipating forest response to climate change can be highly problematic when predictions are made for climatic conditions beyond the data used to fit a model^{38,39}. Here, projected responses to temperature increases of 1-3°C remain within the observed range of T_{MAX} across the entire boreal study area for all species (Table S3). For a 4°C increase in T_{MAX} , the fraction remains above 97% for all species except jack pine (>85%). A 4°C increase in T_{MAX} in these stands is $0.2 \pm 0.2^\circ\text{C}$ (mean \pm SD) above observed temperatures. Additional analyses were performed excluding these areas from future growth projections, with marginal effects on the general trends reported here, suggesting that our projections for the boreal zone are robust. »*

REVIEWER 2 - It is not entirely clear to me why the authors used CMI as opposed to precipitation to represent moisture in their model. They present the growth projections under climate change (Fig. 3 and related text) in terms of changes in precipitation (i.e., -5% and +15%), presumably because this is easier for readers to relate to than changes in CMI. I'm wondering if they explored the possibility of using precipitation directly in their models; this would also have the advantage of being more independent from temperature than CMI (which is calculated using temperature variables).

AUTHORS – Although we agree with the reviewer that precipitation is less correlated with temperature, our current variable to estimate growing season moisture, CMI, is nevertheless poorly correlated with annual Tmax across our study sites ($R^2=0.06$). This low correlation is probably due to the fact that the algorithm used to estimate PET is a simplified Penman-

Monteith, which relies more on radiation than temperature to estimate PET (see ref 40 in manuscript).

In addition to this low correlation, we chose to use CMI and Tmax to represent climate variables as they illustrate climate change impacts which may have opposite effects on boreal tree growth. Based on current knowledge (Price et al. 2013), projected increases in temperature may favor growth by extending the length of the growing season, while warming-induced water deficits during the growing season are increasingly well documented (see refs 8-11 in manuscript).

We have added a justification for the use of these two variables in the Methods section of the manuscript:

(P16 L348): « Average summer (CMI_{SUMMER} ; June to August) and growing season (CMI_{GS} ; May to September) climate moisture indices were used to assess the documented warm-season water constraint on growth⁸⁻¹¹, while average mean (T_{MEAN}) and maximum (T_{MAX}) daily temperature estimates were used to assess the control exerted by annual temperature on growth². Across the study area, CMI_{GS} is poorly relative to T_{MAX} ($R^2 = 0.06$). »

REVIEWER 2 - I see only one mention in the ms of population-level variation in climate change response (Lines 236-239) – with the assumption here being that populations are going to respond consistently across the range of the species. I’m not opposed to this, but this is a key assumption that probably warrants further elaboration – at least recognizing the significant work done in this area vis-à-vis provenance trials, genecology, etc.

AUTHORS – We agree with the reviewer and have added additional references in the text to significant work regarding population-level variations in climate sensitivity in boreal forests of Canada:

(P12 L248): « However, our space-for-time modelling approach assumes that species responses to climatic gradients across space reflect their future local response to climate change over time. Our current knowledge of boreal population-level variations in climate change response remains limited⁴¹, despite significant advances from provenance trials⁴²⁻⁴³ or genecology studies⁴⁴⁻⁴⁵. »

REVIEWER - Specific Comments

REVIEWER 2 - 1. The amount of deviance explained is (unnecessarily) repeated in lines 69 and 71.

AUTHORS – Agreed. We have removed the repeated values.

REVIEWER 2 - 2. Lines 84-87; may be worth mentioning that the linear relationships between growth and climate for certain species are surely due to a lack of samples from one end of the climatic gradient.

AUTHORS – We agree:

(P4 L92): « *Such linear relationships between growth and climate are probably due to a lack of samples from the warm end of the climatic gradient.* »

REVIEWER 2 - 3. Line 137; replace ‘from and’ with ‘from’.

AUTHORS – The sentence was corrected.

REVIEWER 2 - 4. Lines 134-144; when discussing impacts of precipitation on future tree growth it may be worth reminding readers that this study is examining responses to changes in average (multi-decadal) precipitation conditions. Extreme events (droughts and floods), which may increase with climate change and which are not captured in multi-decadal averages, could also have significant impacts on forest growth and mortality.

AUTHORS – We agree with the reviewer that this is an important point. We now mention this at the beginning of the results on future growth modeling.

(P7 L125) « *It should be noted that climate anomalies such as drought, which may increase with climate change, are not captured in multi-decadal average changes presented here but could also have significant impacts on forest growth and mortality.* »

REVIEWER 2 - 5. Line 235; consider removing the word ‘somehow’ from this sentence.

AUTHORS – ‘Somehow’ was removed from the sentence.

Reviewer 3

REVIEWER 3 - In this study the authors apply growth models to the 7 dominant tree species in the eastern Canada boreal region. They found that climate sensitivity was species specific and the dominant climatic limitations varied with latitude. The authors identify climatic thresholds in order to predict future trajectories of this ecosystem in response to climate change. I feel that this work is very interesting and an important contribution to our understanding of the future of the boreal ecosystem. The manuscript was very well written and I have a few minor comments listed below. The only substantive comments I have are in regards to the models. I think the use of a general additive model is great, but my concern lies in the exclusion of random effects. In addition, I feel that clarification of the simulated growth model in regards to the inclusion of CMI is needed (see comments below).

REVIEWER 3 - L 33 – delete increasingly

AUTHORS – The word was removed.

REVIEWER 3 - L36 – 39. This sentence has a lot of information. Perhaps start a new sentence after “warming”.

AUTHORS – The sentence was split into two:

(P1 L36) « *Eastern North America (ENA) is projected to be the planet’s only boreal region with sufficient precipitation to cancel out increases in evapotranspiration associated with future warming¹⁵. However, we still lack accurate estimates of species climatic thresholds in order to predict future trajectories for this ecosystem¹⁶.* »

REVIEWER 3 - L52 – what are the 7 species?

AUTHORS – We have added the names of the seven species in the sentence:

(P2 L51) « *Here, we aimed to model sensitivity to current climate of the seven most abundant boreal tree species in Eastern Canada (black spruce, *Picea mariana*, white spruce, *P. glauca*, balsam fir, *Abies balsamea*, Jack pine, *Pinus banksiana*, Aspen, *Populus tremuloides*, white birch, *Betula papyrifera* and Larch, *Larix laricina*) and to project potential changes in growth to increasing temperature and changes in precipitation.* »

REVIEWER 3 - L74 – P and PET have not been defined above – reference the reader to the methods?

AUTHORS – We have changed the sentence to clarify the terms and added a reference to the Methods section:

(P3 L76) « *Nonetheless, growth responded strongly to temperature, available water (CMI, measured as growing season precipitation minus potential evapotranspiration, see Methods) and the interaction of the two ($P < 0.01$; Table 1).* »

REVIEWER 3 - L75 – interesting that larch is insensitive to climate and competition – I wonder what is limiting larch growth? Are these results in supplementary?

AUTHORS – The lack of sensitivity of the species to available water is likely related to its narrow range of distribution along gradients of soil drainage. Indeed, 51% of sampled larch trees used in the current study were found in wetlands (see Fig S1). We have added some information to the main text:

(P4 L80): « *This lack of response to water availability is likely due to the species predominance in wetlands (51% of sampled individuals; Figure S1).* »

REVIEWER 3 - L85 – remove “e.g.”

AUTHORS – It was removed.

REVIEWER 3 - L 118 – 144 – these results were a bit hard to wade through. I know the trends are presented in the figure – but perhaps the numbers could be condensed into a

table to simplify and just highlight the most important trends rather than the exact numbers in the text?

AUTHORS – We agree with the reviewer. We have addressed this issue by adding a table (now Table 2) to the main text summarizing the numbers from Figure 3 and condensed the text to present the main trends:

(P7 L129): « *Conifer growth displays a strong latitudinal response gradient to warming-only scenarios (Figure 3 & Table 2). Under a 4°C increase in T_{MAX} , black spruce and balsam fir changes in growth shift from marginal declines south of 50°N to important gains north of that latitude. Similarly, jack pine growth increases across its range, but gains increase with latitude. South of 50°N, where most white spruce is found, the species growth displays large negative declines. Our projections suggest that growth decline areas expand with warming. For instance, under a 1°C warming, black spruce growth declines are limited to southeastern Quebec and represent 3% of the species range in the boreal zone but expand to 36% with 4°C warming (Figure 3 & Table 2). The proportion of growth decline in white spruce shifts from 29 to 76% with a 1 to 4°C change. Relative to conifers, broadleaf trees undergo modest relative changes in the southern part of the study area. North of 50°N, birch growth increases significantly with warming, while aspen growth remains largely unchanged across the entire boreal zone. Marginal birch growth declines are mainly observed in the southwest, characterized by low snowfall and precipitation, high summer temperature and abundant glaciolacustrine deposits compared to the till-dominated deposits in the remaining territory.*

Variations in future precipitation could have large impacts on boreal tree growth. For balsam fir stands south of 50°N, a 15% increase in growing season precipitation cancels out the average growth decline following a 4°C warming (Figure 3 & Table 2). For the same region, jack pine growth under a 4°C warming increases from 10±20% (-5% precipitation) to 31±20% (+15% precipitation). Interestingly, our projections suggest that under moderate warming, reduced precipitation could be beneficial to the growth of black spruce, white spruce and jack pine in some high-latitude areas. »

REVIEWER 3 - L280 – I’m curious as to why 9 cm DBH was used – based on growth rates it would seem as though a highly productive tree (white spruce) would be much younger than a tree of very low productivity (black spruce) once this 9cm threshold was reached. I assume that the inclusion of age in the models accounts for this – which is written in the equation on L 403 but not in the text preceding it. Furthermore, sampling only these trees creates a bias that you explain nicely in L715-719 but should be referenced in the text.

AUTHORS – To address the reviewer’s comment, we have appended the text describing the growth model (preceding L403) to include age. In addition, we have added a reference in the text referring to the bias description at L715-719.

(P17 L408): « A single GAM was fitted for each species to predict the annual BAI of a tree j in site i as a function of temperature, water availability, competition, tree size and age, snowfall and soil physical environment, assuming a Gaussian distribution of the response variable. »

(P18 L426): « Tree age was included in the growth model to correct for the well-documented sampling bias caused by the absence of old, fast-growing trees and young, slow-growing trees from the tree ring dataset (see Supplementary Text). »

REVIEWER 3- L417 – but you do have uneven sample sizes associated with this spatial structure, correct? If so, I think that the random effect of site should either be included in your models or should at least be tested and then if insignificant removed.

AUTHORS – Following comments from reviewers 1 and 3, we tested for the presence of spatial autocorrelation in the model residuals. If it was the case, the models would require accounting for this spatial autocorrelation. Results from this analysis, now included in the supplementary materials of the manuscript (Fig. S5), clearly demonstrate the absence of significant spatial autocorrelation in the model residuals. Therefore, we have not included a spatial random structure in our models.

We have added this information in the Methods section:

(P18 L431): « Based on the low spatial autocorrelation of the model residuals (Figure S5), no explicit spatial structure was included in the model. »

REVIEWER 3- L 462 - In reading the main text, I was curious as to why CMI was used in the GAM, but precipitation was used in the simulated growth model. This sentence alludes to the inclusion of CMI in the simulated model, but I don't quite understand how – perhaps include the model formula? I think this needs to be expanded/clarified and should probably be included in the main text.

AUTHORS – The water availability index used here (CMI) is calculated as precipitation minus potential evapotranspiration during the growing season. Therefore, simulated growth models use the same input variables as the fitted GAM, although we present changes in precipitation in the simulations (-5%, +15%) to ease understanding. We have clarified the relationship between both climate factors in the main text:

(P3 L76) « Nonetheless, growth responded strongly to mean daily maximum temperature (T_{MAX}), growing season water availability (CMI, measured as growing season precipitation minus potential evapotranspiration, see Methods) and the interaction of the two ($P < 0.01$; Table 1). »

(P6 L115): « We simulated species sensitivity to likely scenarios of climate change according to local site conditions. Given the uncertainty that remains in future climate conditions, we used various combinations of temperature (1-4°C increases in T_{MAX} and associated increase in PET of 43-173 mm) and precipitation changes (-5 to +15% of growing season precipitation) lying within the range of values predicted by GCMs (Figure S4). Future changes in temperature and

associated PET as well as in precipitation were used to calculate new T_{MAX} and CMI estimates that were introduced into the growth models. »

REVIEWERS' COMMENTS:

Reviewer #1 (Remarks to the Author):

The authors have thoroughly answered all request regarding my comments (rev. 1) and have recalculated some of the values, from that point the paper now looks good - I had also a look at their responses to the comments of the other reviewers and I think they have also adequately taken into account their comments (but I would of course leave this to the judgment of my colleagues). I therefore have no further objections against this paper.

Reviewer #2 (Remarks to the Author):

I have read the author's responses to my review and feel they have adequately addressed my concerns. Again, I think this is an excellent paper that is worthy of publication in Nature Communications.

Reviewer #3 confidentially expressed satisfaction with the revisions

Response to the Referees

After reviewing our responses to their earlier comments, the reviewers expressed no issue to address, thus we provide no additional modifications in this regard.

Here are the reviewer comments:

Reviewer #1 (Remarks to the Author): The authors have thoroughly answered all request regarding my comments (rev. 1) and have recalculated some of the values, from that point the paper now looks good - I had also a look at their responses to the comments of the other reviewers and I think they have also adequately taken into account their comments (but I would of course leave this to the judgment of my colleagues). I therefore have no further objections against this paper.

Reviewer #2 (Remarks to the Author): I have read the author's responses to my review and feel they have adequately addressed my concerns. Again, I think this is an excellent paper that is worthy of publication in Nature Communications.

Reviewer #3 confidentially expressed satisfaction with the revisions.